# Identification of an amphipathic peptide sensor of the *Bacillus subtilis* fluid membrane microdomains

Yiping Jiang[1], Xin Dai[1], Mingming Qin[1] & Zhihong Guo [1]

Regions of increased fluidity are newly found bacterial membrane microdomains that are composed of short, unsaturated and branched fatty acyl chains in a fluid and disordered state. Currently, little is known about how proteins are recruited and localized to these membrane domains. Here, we identify a short amphipathic α-peptide in a previously unreported crystal structure and show that it is responsible for peripheral localization of the phosphate acyltransferase PlsX to the fluid microdomains in *Bacillus subtilis*. Mutations disrupting the amphipathic interaction or increasing the nonpolar interaction are found to redistribute the protein to the cytosol or other part of the plasma membrane, causing growth defects. These results reveal a mechanism of peripheral membrane sensing through optimizing nonpolar interaction with the special lipids in the microdomains. This finding shows that the fluid membrane microdomains may take advantage of their unique lipid environment as a means of recruiting and organizing proteins.

---

[1] Shenzhen Research Institute and Department of Chemistry, The Hong Kong University of Science and Technology, Clear Water Bay, Kowloon, Hong Kong SAR, China. Correspondence and requests for materials should be addressed to Z.G. (email: chguo@ust.hk)

Regions of increased fluidity (RIFs) are fluid membrane microdomains recently found in rod-shaped Gram-positive and Gram-negative bacteria[1,2]. They are likely clusters of lipids with short, branched, and/or unsaturated fatty acyl chains in a fluid and disordered state[1]. In *Bacillus subtilis*, these liquid-disordered microdomains are preferentially bound by the lipo-peptide antibiotic daptomycin that causes membrane rigidification and detachment of peripheral proteins to affect cell wall synthesis[3]. Like the well-known liquid-ordered lipid rafts in both prokaryotic and eukaryotic cells but via an opposite physico-chemical principle[4–9], RIFs are expected to be a platform for organization of proteins to play important physiological roles, including their revealed involvement in the cell wall synthesis. However, very little is known about how proteins are recruited to these membrane microdomains.

The phosphate acyltransferase PlsX is one of the few proteins known to localize to RIFs in *B. subtilis*[3], which are present in a punctate pattern and stained with a high specificity with the lipid analog indocarbocyanine dye DiIC12[1,2]. This enzyme catalyzes the first chemical step in phospholipid biosynthesis by converting acyl–acyl carrier protein (ACP) and phosphate into acyl phosphate and is essential in a large subset of bacteria[10,11]. Its RIFs localization is consistent with its punctate membrane distribution in fixed cells[11] or in live cells[12]. In the latter study, PlsX was also shown to be a cell division protein by also localizing to the septal membrane at the midcell and by interacting with FtsA and other division proteins of the divisome. However, the cell division role is not supported in a more recent study[13], which also notes that the punctate membrane foci of PlsX are formed in the early log phase when the bacterial cells grow at a high rate.

PlsX is an obvious peripheral membrane protein because it was over-expressed and purified as a cytosol protein[10] and yet was consistently observed to associate with the membrane[3,11–13]. However, it is not known whether the protein is tethered to the membrane by direct interaction or indirect interaction with other proteins. This problem is not solved by the PlsX crystal structures determined for the orthologues from *B. subtilis* (PDB: 1VI1[14]) and *Enterococcus faecalis* (PDB: 1U7N[15]). Both crystal structures feature PlsX as a functional dimer in a mushroom architecture with a long stem, which is a four-helix bundle with each subunit contributing two helices. At the dimeric interface, two large symmetric cavities were found to extend from the buried end of the four-helix bundle to the protein surface, which were suspected to be the enzyme-active sites. Despite these functional insights, no clue is available from the crystal structures on how PlsX is localized to the RIFs.

To understand the structural basis of the RIFs localization, we crystallized *B. subtilis* PlsX again in the presence of a product analog and obtained a structure, which contains a short amphipathic α-peptide likely involved in the subcellular localization. By mutating the interfacial residues of the amphipathic peptide and determining their effects on the subcellular localization, we showed that this α-peptide is indeed responsible for the RIFs localization through direct interaction with the membrane. In addition, we also showed that this peptide-mediated RIFs localization is important for the cell growth in a functional assay. Moreover, additional mutational analysis of the amphipathic α-peptide was performed to shed light on the mechanism of its recognition of the fluid membrane microdomains.

## Results

### Crystal structure. 
*B. subtilis* PlsX was crystallized in the presence of the product analog palmitoyl phosphoramide and the crystal structure was determined at 2.30 Å resolution. However, the product analog was not found in the solved structure, in which

the asymmetric unit contains two largely symmetric subunits in a mushroom architecture with a four-helix bundle stem formed at the dimeric interface (Fig. 1a). The structure is closely similar to the previously solved structure of the same protein with a root-mean-squared deviation of 0.56 Å over all comparable $C_\alpha$ carbon atoms, forming two symmetric large cavities suspected to be the active sites at the dimeric interface (circled in Fig. 1a). The only obvious difference with the previous structure lies in a 13-residue fragment (residues 250–262) at the exposed end of the helix bundle stem (Fig. 1b). This fragment is fully ordered with good electron density in the structure, while it is largely disordered in the previously determined structure of the same protein[14]. In the crystal structure of *E. faecalis* PlsX (1U7N[15]), most residues of this fragment form part of the α-helices in the helix bundle stem with four disordered residues in the middle, which correspond to residues 253–256 in *B. subtilis* PlsX.

The 13-residue fragment is structurally different in the two subunits of the dimeric structure. It is comprised of a 9-residue (residues 253–261) α-helix and a 2-residue loop in chain B and a largely random coil loop in chain A (Fig. 1c). Interestingly, the 9-residue α-helical peptide in chain B is amphipathic with Thr253, Thr255, and Lys257 at the interface and Leu254, Leu258, and Ala261 forming the nonpolar surface together with the hydrophobic part of the interfacial residues (Fig. 1d). Ala259 and Ala260 are also close to the polar–nonpolar interface without a polar functional group like other interfacial residues. The N-terminus of the α-peptide is connected to the end of one vertical helix through the 2-residue loop (Thr251 and Ser252) and its C-terminus is connected to another vertical helix of the four-helix bundle stem via a one-residue sharp turn (Val262). This amphipathic α-peptide was formed in the crystal likely due to the presence of the product analog, palmitoyl phosphoramide, which is an anionic detergent by nature. Its absence in previously determined crystal structures might be a result of the lack of a similar detergent in the crystallization buffer[14,15].

Since amphipathic α-peptides are well known to recognize various membrane features for peripheral localization to biological membranes[16–24], the identified 9-residue amphipathic α-peptide is expected to be involved in the subcellular localization of PlsX to RIFs. Nonetheless, a monomeric green fluorescent protein ($GFP_m$) with the peptide at its N-terminus failed to attach to the RIFs, suggesting that a single peptide is unable to localize to the fluid membrane microdomains (Supplementary Fig. 1). In consideration of the overall symmetry of the dimeric structure, this short amphipathic α-peptide is also expected to be formed in the other protomer under physiological conditions. However, repetitive crystallization attempts failed to observe the two expected amphipathic α-peptides in an alternate structure in the presence of other detergents or palmitoyl phosphoramide at an increased concentration.

**Amphipathicity required for the RIFs localization.** To investigate the role of the identified amphipathic α-peptide in its RIFs localization, PlsX was fused to $GFP_m$ at the N-terminus and induced for expression by xylose from the *amyE* locus in *B. subtilis* in the presence of the untagged PlsX expressed from its native locus exactly as previously reported[3]. RIFs puncta were readily detected on bacterial cells at the early log phase by staining with the DiIC12 reporter dye and optimally co-localized with the expressed $GFP_m$-PlsX fusion protein in widefield fluorescent microscopy (Fig. 2a). From the confocal microscope image (Fig. 3), it is obvious that a majority of the fusion protein is associated with the DiIC12-stained RIFs with negligible amount in the cytosol. To disrupt the amphipathicity of the identified α-peptide, the polar interfacial residues Thr253, Thr255, and Lys257 were mutated individually to alanine and the resulting

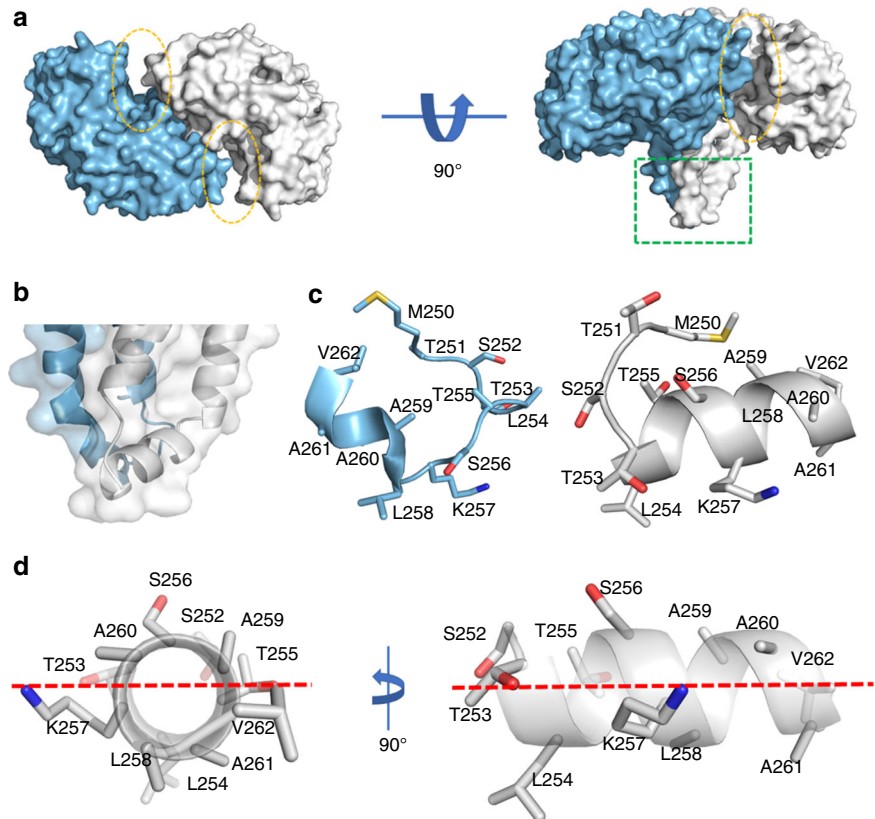

**Fig. 1** Amphipathic α-peptide in the crystal structure of *Bacillus subtilis* PlsX. **a** Two views of the PlsX dimer. The structure is colored according to chains in surface presentation with the exposed end of the four-helix bundle highlighted by a green square; potential active sites are denoted by yellow oval circles. **b** The secondary structure of the exposed end of the four-helix bundle. The part in **a** highlighted in the green square is presented. **c** The 13-residue fragments at the exposed end of the four-helix bundle. They are colored as in **b** with their backbones in cartoon and their side chains in sticks. **d** Two views of the 9-residue amphipathic α-peptide in chain B. Red dashed lines indicate the polar–nonpolar interface; Met250 and Val262 are connecting residues and are not part of the α-peptide; the Leu258 side chain is partly ordered

mutants were not localized to RIFs (Fig. 2a) but were mostly found in cytosol (Fig. 3). Noticeably, no free GFP$_m$ was detected in the cells (Supplementary Fig. 2) and thus the observed delocalization was not due to degradation of the mutant proteins. In comparison, similar alanine mutation of the non-interfacial Ser256 or Thr251 in the 2-residue connecting loop (Fig. 1c) had no effect on the subcellular localization (Fig. 2a). In addition, delocalization of the fusion protein to cytosol was also observed when a serine was introduced to the position of Leu258 on the nonpolar face of the amphipathic α-peptide (Figs. 2a and 3). However, when L254E, L258E, or A261D was introduced to the nonpolar face, the RIFs localization was not changed (Fig. 2a). This unexpected result is likely due to snorkeling of the introduced side-chain carboxylate to the polar–nonpolar interface, which is allowed by the close proximity of the mutated residues to the interface. To test this, L254D mutant was made with a shorter side chain compared to L254E and was indeed found to delocalize to cytosol like other amphipathicity-disrupted mutants (Fig. 2a). Moreover, recombinant PlsX was found to directly bind to protein-free total lipids from *B. subtilis* and the K257A mutation was found to weaken this interaction (Supplementary Fig. 4). Taken together, these results provide strong evidence that PlsX interacts directly with membrane for its peripheral association with RIFs and that the amphipathic α-peptide is responsible for this subcellular localization.

**Hydrophobic interaction in the RIFs localization**. To understand how the RIFs localization is affected by the hydrophobic interaction of the amphipathic α-peptide with the membrane,

Leu254 and Leu258 on the nonpolar face were mutated separately to alanine. The resulting mutant fusion proteins were both delocalized into the cytosol (Figs. 2c and 3). For comparison, the subcellular localization of the fusion protein showed no change for similar alanine mutation of Met250 and Val262 at the two ends of the 13-residue fragment (Fig. 2c). The unchanged RIFs localization of the V262A-containing mutant strongly suggests that the Val262 side chain makes no contact with the membrane although this residue is spatially close to Ala261 and makes the sharp transition from the amphipathic α-peptide to the vertical helix (Fig. 1c, d).

To explore the effect of increased hydrophobic interaction, the interfacial residue Ala259 was mutated to threonine while Thr253 and Thr255 were replaced with tryptophan, a snorkeling residue with the largest hydrophobic moiety for interaction with acyl groups of the membrane lipids[25,26]. Under widefield fluorescence microscopy, the A259T-containing mutant was not visible due to poor expression (Supplementary Fig. 2), whereas both T253W and T255W-containing mutants are mostly distributed throughout the whole cell but remain partially enriched on the DiIC12-stained RIFs (Fig. 2b). Under confocal fluorescence microscopy, a majority of the T253W-containing mutant was delocalized to the cytosol but the T255W-containing mutant remains associated with the lateral membrane and the midcell region (Fig. 3). In corroboration of this result, the GFP$_m$-T255W protein was found to be associated with the membrane at a level similar to the wild-type GFP$_m$-PlsX but higher than GFP$_m$-T257A (Supplementary Fig. 5), which was mainly delocalized to cytosol. On the other

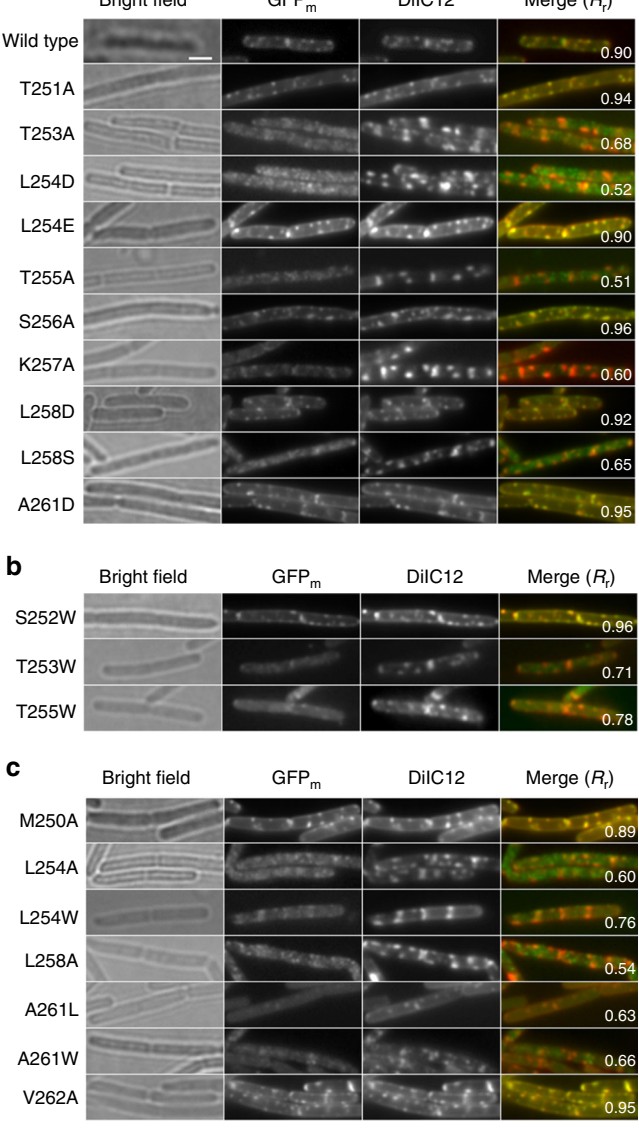

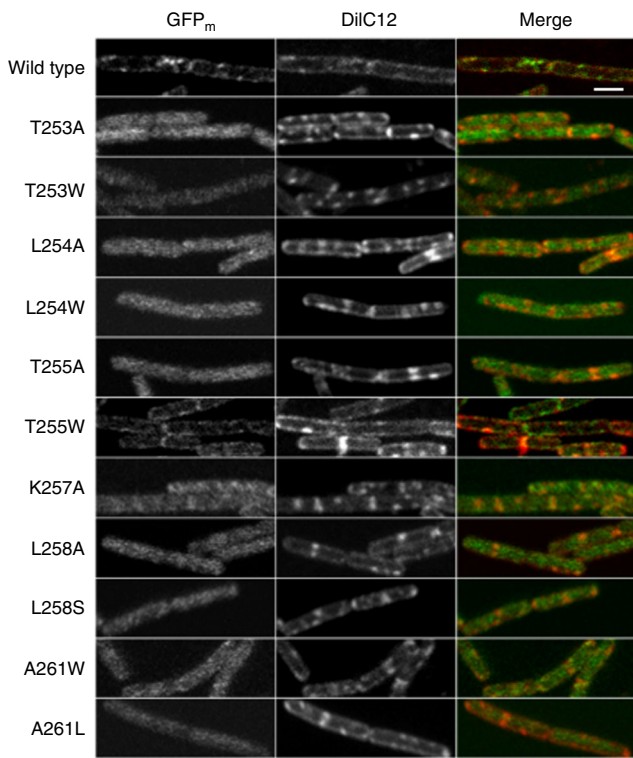

**Fig. 2** Co-localization of GFP$_m$-PlsX and its mutants with the DiIC12-stained RIFs by widefield fluorescence microscopy. **a** Effect of mutations disrupting amphipathicity of the α-peptide. **b** Effect of mutations increasing the hydrophobic interaction at the polar–nonpolar interface. **c** Effect of mutations varying the hydrophilicity of the residues on the non-polar side of the amphipathic α-peptide. Yellow color in the merged image indicates co-localization of the fluorescent labels DiIC12 and GFP$_m$, which emit red and green lights, respectively. The scale bar is 2 μm and the same for all images, which are cut from a larger area that include at least 10 cells as shown in Supplementary Fig. 3. The larger images are used to calculate the Pearson's correlation coefficient ($R_r$) that is an indicator of the extent of co-localization and is provided in the merged images

**Fig. 3** Subcellular localization of GFP$_m$-PlsX and its mutants by confocal microscopy. All the mutants are delocalized from RIFs. The green (GFP$_m$) and red (DiIC12) images of the cell slice (set to 0.5 μm) are not perfectly superimposable due to a small change of the cell slice thickness caused by variation in the wavelength of the input light. The scale bar is 2 μm and the same for all images

was also found to be partly associated with the RIFs foci with the highest proportion for the L254W-containing mutant, the second for the A261L-containing mutant, and a negligible amount for the A261W-containing mutant (Fig. 2c). Noticeably, all the mutant proteins were expressed as a full-length fusion with GFP$_m$ (Supplementary Fig. 2) and thus their delocalization to the cytosol was not due to free green fluorescent protein released from their degradation.

**Effect of the RIFs localization on cell growth**. Detachment of PlsX and other peripheral proteins from the RIFs has been suggested to be a major contributor to the daptomycin antibacterial effect of inhibiting synthesis of phospholipids and the cell wall[3]. However, this proposed role of the subcellular localization is complicated by the multiple membrane-disrupting effects of daptomycin[27,28]. To determine the putative functional effect of the RIFs localization, we used a mutant *plsX* gene containing the S256A, K257A, or T255W point mutation to replace the wild-type gene and expressed the mutant proteins under the native promoter. In the meantime, the wild-type *plsX* gene was used as a control in the gene replacement to avoid polar effect. The resulting *B. subtilis* strains were readily obtained. As shown in Fig. 4, the strains with the gene replacement with the wild-type *plsX* gene or its S256A mutant with unchanged RIFs localization are almost indistinguishable from the wild-type *B. subtilis* strain 168 in their vegetative growth curves except a higher plateau for the latter, suggesting a mild polar effect for the gene insertion. In contrast, the strain expressing either the K257A or T255W mutant shows a similar two-hour delay in transition to the exponential phase and a plateau lower than the S256A mutant.

hand, several point mutations were introduced to the nonpolar surface of the amphipathic α-peptide to increase the hydrophobic interaction with the membrane, including A261L, A261F, A261W, and L254W. The A261F mutant was not substantially expressed and was invisible under a widefield fluorescence microscope, while all other mutants were expressed at a level comparable to the wild-type protein (Supplementary Fig. 2) with a different pattern of subcellular localization. A majority of each detectable mutant protein was found in the cytosol and evenly distributed throughout the cell (Fig. 2c), but the fusion protein

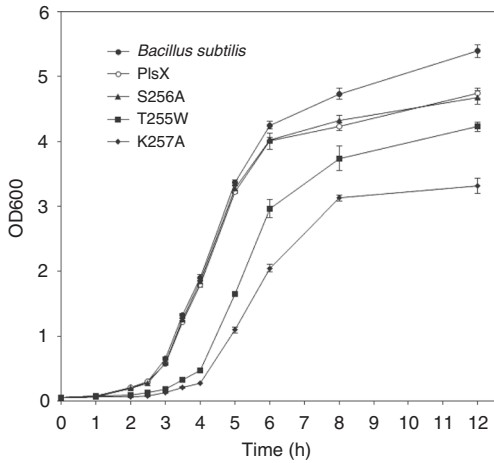

**Fig. 4** Growth defect caused by mutations disrupting the RIFs localization. Overnight culture of each cell strain was diluted to the same density and grown in triplicate in Luria-Bertani medium at 37 °C under identical conditions. 'Bacillus subtilis' denotes the wild type Bacillus subtilis strain 168 without gene replacement; 'PlsX' denotes the same cell strain with its plsX gene replaced by the wild-type plsX in pSG1151; while 'S256A, T255W, K257A' denote the cell strains with the gene replacement by the mutated plsX gene carrying the indicated mutation in pSG1151

This growth impairment is greater for the K257A mutant, which is mostly delocalized to the cytosol, than the T255W mutant, which is mostly redistributed to the non-RIFs membrane, due to the apparent different plateau level in their growth curves. Since the point mutations are very far from the suspected active site (Fig. 1a) and should not affect the catalytic activity, this growth impairment provides unambiguous evidence for a crucial role for the subcellular localization in the physiological function of PlsX. This is further supported by the unaffected catalytic activity of the pure recombinant K257A protein in comparison to the non-mutated PlsX (Supplementary Fig. 6).

## Discussion

Amphipathic α-peptides are structural motifs commonly found in many proteins for specific recognition of and binding to biological membranes according to their structural features, such as curvature, lipid composition, and charges[16–24]. In this study, we have identified a amphipathic peptide in the phosphate acyltransferase PlsX which specifically recognizes and binds membrane regions according to their fluidity, a previously unreported membrane feature. It locates in the middle of the protein and works as a dimer that caps the exposed end of the four-helix bundle stem in the enzyme's mushroom-like structure. This peptide contains nine residues in two and a half α-helical turns and is too short to be predicted from its amino acid sequence by HeliQuest[29]. The small size is comparable to the membrane targeting peptide of MinD, which also works as a dimer to involve in placement of the bacterial cell division site[30]. Importantly, the peripheral localization mediated by this short amphipathic peptide is shown to be important for the physiological function of PlsX by the growth impairment caused by its putative disruption, which also shows the importance of RIFs in phospholipid biosynthesis.

The two-peptide working model of the identified amphipathic α-peptide is consistent with several lines of experimental observation. Besides being in line with the overall symmetry of the PlsX crystal structure (Fig. 1a), this model is supported by the failure of localization of the monomeric amphipathic α-peptide in fusion with GFP_m to RIFs (Supplementary Fig. 1). In addition, this model is able to explain the subcellular localization of many GFP_m-PlsX

mutants, such as K257A, T253A, and T255A, which are mostly delocalized to the cytosol or other parts of cell membrane but yet retain residual co-localization with RIFs (Figs. 2 and 3). Although this partial delocalization from RIFs can be explained by reduced strength in the interaction of the mutated peptide with membrane lipids, it is better understood on the basis of the model that allows formation of a hybrid dimer composed of a wild type PlsX monomer without fusion and a PlsX mutant in fusion with GFP_m. Such hybrid dimers could be formed from subunit swapping between a wild-type PlsX dimer and a GFP_m-PlsX mutant dimer that are both produced with the former expressed at a higher amount according to a previous estimation in a closely similar expression system[12]. These hybrid dimers are less affected by the point mutations in their interaction with membrane lipids in comparison to the fluorescent homodimers and may still be able to localize to RIFs. Therefore, the residual RIFs localization of the mutants is likely an experimental indicator of the two-peptide working model for the peripheral recognition and specific localization to the fluid membrane microdomains.

Specific localization of RIFs mediated by the short symmetric amphipathic α-peptides is a result of their balanced interaction with the polar head groups and the nonpolar acyl chains of the lipids in the fluid microdomains. The polar interaction is apparently kept to a minimal strength barely enough for adhesion of the short α-peptides to RIFs, because their membrane association is lost when a pair of hydrogen bonds are removed by mutating Thr253 or Thr255 into alanine. However, strengthening this interfacial polar interaction appears to be tolerated as seen in the unchanged RIFs localization for proteins containing the A261D, L254E, and L258D mutations (Fig. 2a), in which the sidechain carboxylate snorkels to the polar–nonpolar interface to strengthen the polar interaction with the head groups of membrane lipids. This tolerance may greatly increase the structural diversity for the RIFs-recognizing amphipathic α-peptides, allowing them to have a polar residue on its hydrophobic surface.

In contrast, the nonpolar interaction is more tightly controlled in the RIFs localization of the symmetric amphipathic α-peptides. A decrease in this interaction is not tolerated as seen from delocalization of the fusion protein to cytosol when a leucine residue (Leu254 or Leu258) on the nonpolar face is mutated to alanine (Figs. 2 and 3). An increase in this nonpolar interaction is also not tolerated since the protein is at least partially delocalized from the RIFs foci, no matter whether the increase happens at the interface (for T253W and T255W) or the nonpolar surface of the amphipathic peptide (for A261L, A261W, and L254W). Interestingly, the extent of delocalization from the RIFs appears to correlate with the magnitude of hydrophobicity increase on the nonpolar surface. The fusion protein is partially retained on the RIFs when the hydrophobicity increase is moderate as seen for the A261L and L254W mutants, whilst the delocalization is complete when the hydrophobicity increase is bigger in the A261W mutant (Figs. 2 and 3). In comparison, the interfacial mutations T253W and T255W should cause a similar hydrophobicity increase as the A261W mutation, but these mutants exhibit a different subcellular distribution with partial RIFs co-localization, which is likely due to the difference in the site of the hydrophobicity increase. More interestingly, the T255W mutant is mostly re-distributed from the RIFs to other parts of the membrane because it is still associated with the membrane in confocal fluorescence microscopy (Fig. 3 and Supplementary Fig. 5), while the T253W mutant is mainly delocalized to the cytosol (Fig. 3). One possible explanation for this difference is that Trp255 (in the T255W mutant) is located on the side facing the amphipathic α-peptide of the other subunit in the functional dimer (Fig. 1b, c) to allow the two symmetric tryptophan residues to interact with each other while Trp253 (in the T253W mutant) is located on the opposite side of the

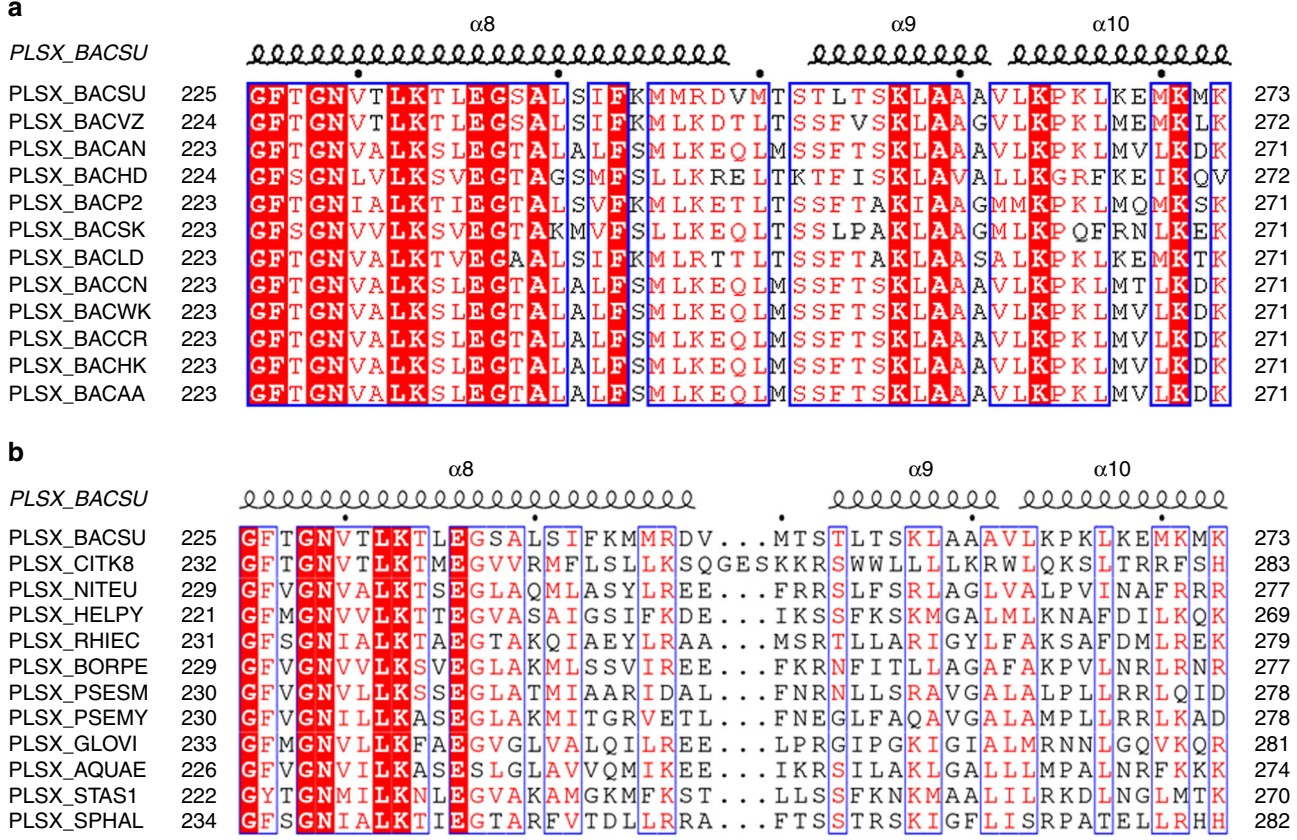

**Fig. 5 Sequence conservation of the PlsX amphipathic α-peptide. a** Alignment of the orthologues from the *Bacillus* genus. PlsX sequences were reviewed, non-redundant sequences deposited in the UniProt database. **b** Alignment of the orthologues from different bacterial genera. PlsX sequences were chosen from 95 reviewed PlsX sequences filtered at 50% sequence identity from the UniProt database, according to the number of PlsX orthologues they represent. The sequences are named directly by their UniProtKB mnemonic identifiers and aligned with the *Bacillus subtilis* PlsX structure determined in the current study (PDB ID: 6A1K). α9 is the RIFs-recognizing amphipathic α-peptide

α-peptide and unable to interact similarly. However, how this probable difference in the side chain interaction leads to different subcellular distribution is not clear. Nevertheless, the RIFs delocalization effect of the hydrophobicity-increasing mutations strongly support that the amphipathic α-peptide keeps its hydrophobicity in a narrow range to achieve specific localization to the fluid membrane microdomains.

The limitations on the nonpolar face of the amphipathic α-peptide are apparently imposed by the RIFs lipids to maximize the preferential interaction to achieve the subcellular localization. Although RIFs are suggested to comprise of lipids with short, branched and unsaturated fatty acyl chains[1,3], their exact lipid composition is not known. This unknown lipid composition is further complicated by the fact that phospholipids are formed mainly from branched *iso-* and *anteiso-*fatty acids with 15 or 17 carbon atoms in *B. subtilis* and that they are negligibly unsaturated under normal conditions[31,32], despite the presence of a desaturase system to respond to cold shock[33–35]. As a result, it is not clear how the revealed features of the short amphipathic α-peptides lead to preferential interaction with RIFs. Nevertheless, the revealed limitations on the amphipathic α-peptide are consistent with its high level of conservation in the *Bacillus* genus in a sequence alignment (Fig. 5a). Among the 12 non-redundant sequences, the interfacial lysine is strictly conserved while Thr253 is replaced by a serine and Leu254 is replaced by a phenylalanine with similar hydrophobicity in most sequences. Interestingly, Thr253 is replaced by proline in *Bacillus clausii* (PLSX_BACSK) and by a bulky hydrophobic residue, valine or isoleucine, in

*Bacillus velezensis* (PLSX_BACVZ) and *Bacillus halodurans* (PLSX_BACHD). In addition, Ala261 is replaced by a serine in *Bacillus licheniformis* (PLSX_BACLD) and a glycine in three other strains. These mild alterations suggest that the amphipathic α-peptide-mediated localization mechanism is flexible and that it may have to slightly change in response to the different lipid environment in RIFs of different bacterial strains. Alternatively, the amphipathic α-peptide may take a slightly different structural form and make corresponding changes in its compositional residues to keep the hydrophobic interaction with the RIF lipids at an optimal level.

When compared with the PlsX orthologues from other bacterial genera, the amino acid sequence of the identified amphipathic α-peptide is not well conserved despite the high sequence conservation in other structural motifs (Fig. 5b). This lack of conservation may be due to the absence of RIFs in other bacteria, which is however unlikely because of the finding of these membrane microdomains in *Escherichia coli*[2]. Alternatively, it can also be explained if PlsX is not specifically localized to RIFs in most other bacterial genera. More likely, it may be due to the difference in the lipid composition of RIFs in different bacterial genera. As mentioned earlier, the fatty acyl composition is distinct in the *Bacillus* genus and very different from that of all other bacterial genera[31,32], raising the possibility that the amphipathic α-peptide has to change in order to adapt to the altered lipid composition of RIFs in other bacteria. Further investigation is needed to fully understand the low sequence conservation for the RIFs-recognizing motif outside the *Bacillus* genus.

In summary, we have found a short amphipathic α-peptide in the phosphate acyltransferase PlsX, which is responsible for sensing RIFs for specific localization in *B. subtilis*. Through mutational analysis, the amphipathic peptide is found to recognize the fluid membrane microdomains through limiting its hydrophobic interaction with the membrane in a narrow range and close to the minimum level. This requirement of strictly restricted hydrophobic interaction may underlie the small size of the peptide and the minimum level of interfacial polar interaction also found through the mutational analysis. This stringent restriction is apparently required for optimal interaction of the amphipathic α-peptide with the special lipids in RIFs, suggesting that the fluid membrane microdomains may take advantage of their unique lipid environment as a means of recruiting and organizing proteins. It would be interesting to see whether similar restriction of hydrophobic interaction plays a role in the RIFs localization of transmembrane proteins or other monotopic proteins.

## Methods

**Synthesis of 1-oxohexadecylphosphoramidic acid**. The product analog was synthesized using a reported method[36]. Briefly, 3 mmol palmitoyl chloride and 12 mmol diethyl phosphoramidate were dissolved in 12 ml dry tetrahydrofuran and cooled down to −78 °C. 12 mmol n-BuLi was added dropwise and the reaction was stirred overnight before being quenched by 1 ml acetic acid. The mixture was washed three times with brine and dried over anhydrous $Na_2SO_4$. Then the intermediate product was further purified by silica gel column chromatography using ethyl acetate/hexane at 1:3 as eluent and the solvent was removed to yield diethyl 1-oxohexadecyl phosphoramidate as a white powder. In the second step, 50 mg 1-oxohexadecyl phosphoramidate was dissolved in 2 ml $CH_2Cl_2$, to which 200 μl of trimethylsilyl bromide was then added dropwise. Subsequently the reaction mixture was stirred at room temperature overnight. After trimethylsilyl bromide and $CH_2Cl_2$ were removed in vacuo, 1 ml 95% ethanol was added to the residue and the mixture was stirred for 1.5 h at room temperature. Then ethanol was removed and 8 ml ether was added to wash the white solid product C16NP, which was dried in vacuo. The final isolated yield of 1-oxohexadecylphosphoramidic acid was 7%. $^1$H NMR ($d_6$-dimethylsulfoxide, 400 MHz): δ 0.86 (t, $J = 6.4$ Hz,3H), 1.24 (s, 24H), 1.46 (m, 2H), 2.17 (m, 2H), 8.95 (d, $J = 8.8$ Hz, 1H).

**PlsX expression and purification**. The *plsX* gene was amplified from the *B. subtilis* genomic DNA using the primers PlsX-for and PlsX-rev listed in Supplementary Table 1 and integrated into the pET28a vector (Novagen) between NcoI and XhoI. The inserted gene was sequenced in full-length by Beijing Genomics Institute (BGI, Shenzhen, China) to ensure that no mutation was introduced. The recombinant plasmid was transformed into *E. coli* C43 (DE3) (Lucigen) and the resulting cells were incubated overnight at 37 °C on Luria-Bertani broth (LB) agar plates supplemented with 50 μg/ml kanamycin. For expression of the protein with a C-terminal hexahistidine tag, a single colony of the recombinant cells was picked to inoculate 10 ml LB containing 50 μg/ml kanamycin and grown overnight at 37 °C with shaking at 250 rpm to give a starter culture. The starter culture was used to inoculate 4 L LB medium supplemented with 50 μg/ml kanamycin and the cells were grown at 37 °C with shaking at 250 rpm until OD$_{600}$ reached 0.4–0.6. Then isopropyl-β-D-thiogalactopyranoside was added to a final concentration of 0.2 mM to induce PlsX expression at 18 °C for 20 h.

The liquid culture was cooled on ice and the cells were harvested by centrifugation and washed once with ice-cold TE buffer (10 mM Tris–HCl, pH 8.0, 1 mM EDTA). The harvested cells were then re-suspended in the start buffer A (20 mM Tris–HCl, pH 8.0, 500 mM NaCl, 20 mM imidazole, 10% glycerol) and lysed by sonication. The cell lysate was centrifuged at 30,000 × g for 15 min to remove cell debris and the supernatant was loaded onto a 5 ml HisTrap HP column (GE Healthcare) that was pre-equilibrated with the start buffer. The column was washed by 50 ml start buffer and weakly bound impurities were removed by washing with 50 ml of 20 mM Tris–HCl buffer (pH 8.0) containing 500 mM NaCl, 100 mM imidazole, and 10% glycerol. The protein PlsX was eluted in ~20 ml 20 mM Tris–HCl buffer (pH 8.0) containing 500 mM NaCl, 300 mM imidazole, and 10% glycerol. The eluted protein solution was concentrated with Millipore YM-30 and further purified by gel filtration using a HiPrep 16/60 Sephacryl S-200 column (GE Healthcare) that was pre-equilibrated with the storage buffer (10 mM HEPES, pH 7.5, 150 mM NaCl, 10% glycerol). The eluted protein was collected and concentrated to 15 mg/ml before crystallization and stored at −20 °C. All steps in this purification were carried out at 4 °C.

**PlsX structure determination and analysis**. Crystallization was carried out using the hanging drop method by mixing a protein solution and reservoir solution at 1:1 ratio under room temperature (22 °C). The protein solution was prepared by dissolving the product analog 1-oxohexadecylphosphoramidic acid in the PlsX solution

### Table 1 Data collection and refinement statistics[a]

| | PlsX (PDB code: 6A1K) |
|---|---|
| **Data collection** | |
| Space group | C222 |
| Cell dimensions | |
| $a$, $b$, $c$ (Å) | 108.52, 144.76, 84.84 |
| $α$, $β$, $γ$ (°) | 90, 90, 90 |
| Resolution (Å) | 33.29–2.30 (2.382–2.30) |
| $R_{merge}$ | 0.100 (0.428) |
| $I/σ(I)$ | 11.5 (3.1) |
| $CC_{1/2}$ | 0.993 (0.929) |
| Completeness (%) | 96.2 (90.9) |
| Redundancy | 6.4 (5.6) |
| **Refinement** | |
| Resolution | 33.29–2.30 |
| No. of reflections | 29,907 |
| $R_{work}/R_{free}$ | 0.2025/0.2379 |
| No. of atoms | 4889 |
| Protein | 4620 |
| Ligand/ion | 10 |
| Water | 259 |
| *B* factors | 31.67 |
| Protein | 31.29 |
| Ligand/ion | 72.95 |
| Water | 36.86 |
| R.m.s. deviations | |
| Bond lengths (Å) | 0.007 |
| Bond angles (°) | 0.96 |

[a]Values in parentheses are for the highest-resolution shell

with a final protein concentration of 2.5–7.5 mg/ml. Rod-like crystals appeared over a week and grown to the full size within 3 weeks in a solution containing 5 mg/ml PlsX, 1 mM 1-oxohexadecylphosphoramidic acid, 0.08 M Tris–HCl (pH 8.0), 0.16 M Li$_2$SO$_4$, 21% PEG 4000, 0.012 mM CYMAL 7 and 5% t-butanol. Crystals were cryoprotected with the mother liquor supplemented with 15% PEG 400 and flash-cooled and stored in liquid nitrogen and diffraction data were collected up to 2.30 Å at 100 K with an ADSC Quantum 315R charge-coupled device detector at beamline BL17U1 and BL19U1 at Shanghai Synchrotron Radiation Facility (SSRF)[37] and processed with iMosflm[38] and Aimless[39] in the CCP4 suite[40]. The structure was solved by molecular replacement by Phaser-MR in PHENIX[41], using the previously solved structure of *B. subtilis* PlsX (PDB ID: 1VI1) as the search model. The structure model was further built using COOT[42] and refined by PHENIX-Refine[43]. Mol-Probity[44] was used to assess the overall quality of the structural model. Data collection and refinement statistics are shown in Table 1 and the structure has been deposited in the Protein Data Bank (PDB ID: 6A1K). In structural analysis, all graphics were generated by PyMOL Version 1.3[45]. Multiple sequence alignment was performed with Clustal Omega[46] for structure-based presentation using ESPript 3.0[47].

**Activity assay and direct lipid binding of PlsX and its K257A mutant**. To introduce the K257A mutation, the plasmid in pET28 used in PlsX expression was used as a template for site-directed mutagenesis using the primers in Supplementary Table 1. The K257A mutant was expressed and purified exactly like the wild-type PlsX as described above. The palmitoyl-ACP substrate was prepared enzymatically from palmitoyl-CoA and *apo*-ACP according to a reported method[48]. The enzymatic activity was determined based on coupling with 5, 5-dithio-bis-(2-nitrobenzoic acid), which reacted with the *holo*-ACP product to yield 2-nitro-5-thiobenzoic acid for UV–Vis measurement at 412 with an extinction coefficient of 14,150 $M^{-1}$ $cm^{-1}$. In our activity assay, a 200 μl reaction mixture contained 1 mM MgCl$_2$, 500 μM 5, 5-dithio-bis-(2-nitrobenzoic acid), 500 μM phosphate, and 10 μM palmitoyl-ACP in 50 mM Tris buffer (pH 7.5). The enzyme was then added to a final concentration of 1 μM and the reaction was monitored for absorbance change at 412 nm a UV–VIS spectrometer (Shimadzu). Assays were performed in triplicate at 25 °C.

The protein-free total lipids were prepared from *B. subtilis* using a reported method[32]. Briefly, *B. subtilis* 168 strain was grown in LB medium under aerobic condition at 37 °C and harvested when OD$_{600}$ reached 0.5. Cells from 2 L liquid culture were combined and washed twice with 1% NaCl solution. Subsequently, the cell pellet was added 8 ml chloroform, 16 ml methanol, and 6 ml water and extracted by vigorous mixing. The resulting mixture was left at 25 °C overnight, followed by the addition of 8 ml chloroform and 8 ml water. The lower phase was collected and the solvent was removed by rotary evaporation to obtain a lipid film. Then the lipid film was resuspended in 2 ml buffer containing 10 mM HEPES

(pH 7.5) and 150 mM NaCl and liposome was prepared by 1 h sonication treatment. Subsequently, wild-type PlsX, K257A and lysozyme were added at a final concentration of 2 mg/ml to equal aliquots of liposome and incubated with at 25 °C for 1 h. The mixtures were then ultracentrifuged at 200,000×g for 1 h to collect the liposome fraction for analysis by SDS–PAGE.

**Expression of GFP$_m$-PlsX and its mutants in *B. subtilis*.** *B. subtilis* 168 strain and the vector pSG1729 for N-terminal fusion of green fluorescent protein (GFP) were acquired from Bacillus Genetic Stock Center (BGSC). The *gfp* gene in the vector was introduced the A206K mutation to avoid dimerization[49], using the QuickChange II XL site-directed mutagenesis kit (Agilent) and the primers GFP-for and GFP-rev (Supplementary Table 1). The coded protein, called monomeric fluorescent protein (GFP$_m$), was then fused to the N-terminus of PlsX by amplifying the *plsX* gene from the genomic DNA using the primers WT-for and WT-rev (Supplementary Table 1) and inserting it into the modified pSG1729 vector between BamHI and XhoI. The GFP$_m$-PlsX fusion protein expressed from the resulting plasmid is the same as the GFP-PlsX protein (labeled as PlsX-GFP) in a previous study[3]. To introduce point mutations into the amphipathic α-peptide of the fusion protein, its gene was mutated using the QuickChange II XL site-directed mutagenesis kit (Agilent) and the primers listed in Supplementary Table 1. For T253W, L254W, A261L, and A261W, a different protocol as reported earlier was used to introduce the mutations[50]. For fusion with the amphipathic α-peptide, the coding sequence of the peptide with the residues 250–262 was inserted to the 3′-end of the green fluorescent protein gene in the vector pSG1729 with or without the A206K mutation between BamHI and XhoI using the primers HX1 and HX2. The inserted genes of the fusion proteins were sequenced in full-length by Beijing Genomics Institute (BGI, Shenzhen, China) to ensure that no unwanted mutation was introduced.

For analysis of the expression level of the fusion protein by Western-blotting, the plasmid carrying its gene was transformed into *B. subtilis* 168 according to a reported method[51]. The transformants were selected overnight on LB agar plates supplemented with 50 μg/ml tryptophan and 100 μg/ml spectinomycin at 37 °C to allow genomic integration of the fusion gene and the spectinomycin adenyltransferase gene cassette at the *amyE* locus. For Western blotting, a single colony was used to inoculate 3 ml LB liquid culture at 37 °C, which was supplemented with 50 μg/ml tryptophan, 0.5% xylose, and 100 μg/ml spectinomycin. When OD$_{600}$ reached 0.3, cells were collected, re-suspended in 0.3 ml buffer containing 25 mM Tris–HCl (pH 7.5) and 150 mM NaCl, and lysed by sonication. Cell lysates were subjected to SDS–polyacrylamide gel electrophoresis and the separated protein bands were transblotted onto a PVDF membrane (Bio-Rad) by electroblotting at 4 °C and a constant voltage of 80 V for 3 h in the buffer containing 25 mM Tris–HCl (pH 8.3), 190 mM glycine, and 20% methanol. The PVDF membrane was subsequently washed five times with the TBST buffer (containing 25 mM Tris–HCl, 150 mM NaCl, and 0.1% Tween-20 at pH 7.5) and incubated with the blocking buffer (TBST buffer with 3% BSA) for 1 h at room temperature. It was then immersed in 0.33 μg/ml GFP-specific monoclonal antibody from mouse (ThermoFisher) in blocking buffer and incubated with swirling for 1.5 h. After washing five times with the TBST buffer, the blot proteins were incubated with the HRP-goat anti-mouse IgG antibody (ThermoFisher) at 0.75 μg/ml in the blocking buffer for 1 h. After washing for another five times using the TBST buffer, the blot was treated with SuperSignal West Pico PLUS Chemiluminescent Substrate (ThermoFisher) and the chemiluminescent signal was detected using the ChemiDoc Touch Imaging System (Bio-Rad). Wild-type *B. subtilis* and the transformed strain (with the modified pSG1729 only) expressing the free GFP$_m$ were used as controls. In addition, expression of the GFP$_m$-PlsX fusion protein was compared with and without the xylose induction. The blotting results are given in Supplementary Fig. 2. Noticeably, the expression level of the free GFP$_m$ was very low while the A259T and A261F mutants of the fusion protein were not detectable under the blotting conditions.

To determine the membrane association of proteins, *B. subtilis* cells expressing GFP$_m$-PlsX, GFP$_m$-T255W, and GFP$_m$-T257K were individually grown at 37 °C as described above to OD$_{600}$ = 0.5. The cells were harvested, resuspended in 10 ml standard PBS buffer (137 mM NaCl, 2.7 mM KCl, 10 mM Na$_2$HPO$_4$, and 1.8 mM KH$_2$PO$_4$, pH 7.4) and lysed by sonication. Cell debris was removed by centrifugation at 30,000×g for 15 min, and the supernatant was ultracentrifuged at 100,000×g for 1 h to separate cell membrane and cytosol. The membrane pellet was then resuspended in PBS buffer supplemented with 1% *n*-dodecyl-β-D-maltopyranoside (DDM). Subsequently, membrane and cytosol fractions were subjected to fluorescence measurement using FlexStation 3 Multi-Mode Microplate Reader (Molecular Devices), GFP fluorescence intensity was measured with excitation at 488 ± 4.5 nm and emission at 507 ± 7.5 nm.

For expression of PlsX or its mutants under the native promoter, the wild-type or mutated *plsX* gene was amplified from plasmids constructed above using primers MT-for and MT-rev (Supplementary Table 1) and inserted into the vector pSG1151 between KpnI and XhoI. All the inserted genes had a stop codon at their 3′-end and their sequence were verified via full-length DNA sequencing by Beijing Genomics Institute (BGI, Shenzhen, China). The resulting plasmids were used to transform *B. subtilis* strain 168 and integration of the cloned gene to the *plsX* locus was selected by plating on an LB agar plate containing 5 μg/ml chloramphenicol and overnight incubation at 37 °C. In the cell growth test, a single colony was picked for each cell strain to inoculate 1 ml LB culture supplemented with 50 μg/ml

tryptophan at 37 °C. The overnight culture was then diluted to a low optical density at 600 nm (OD$_{600}$ = 0.05) to inoculate 3 × 10 ml fresh LB medium in three separate shake bottles at 37 °C. Aliquots of the cultures were withdrawn at different time points to monitor the cell growth with OD$_{600}$.

**Fluorescence imaging.** For fluorescence imaging, a single colony of the recombinant *B. subtilis* cells expressing the GFP$_m$-fused protein was used to inoculate a starter culture in 1 ml LB supplemented with 50 μg/ml tryptophan and 0.5% xylose and grown overnight at 37 °C. Next day, the starter culture was diluted 100-fold into the same medium containing 1% dimethylsulfoxide and 2 μg/ml DiIC12 and grown at 37 °C until OD$_{600}$ reached 0.3. Then cells were collected by centrifugation and washed three times and re-suspended with prewarmed medium without DiIC12 at 37 °C. Before cell immobilization, a piece of 25 μl Gene Frame (ThermoFisher) was adhered onto a glass slide and the well was filled with 1.5% agarose melt in hot LB. After agarose hardened and cooled to room temperature, air chambers were created according to the reported method[52] and 1 μl of re-suspended cells was pipetted onto the agarose film. After the medium dried up due to absorption by the agarose film, the immobilized cells were covered with a coverslip (Marienfeld) and imaged with a Nikon TE2000E-PFS microscope equipped with an Intensilight Epi-fluorescence Illuminator, an Andor EMCCD BOOST camera, an Alpha-PL APO ×100 oil objective, a GFP filter cube, a rhodamine filter cube and NIS element software or a Zeiss Laser Scanning Confocal Microscope (LSM 710) equipped with a PL APO ×63 oil objective, 488 nm Argon laser, 560 nm DPSS laser source, and Zen 2009 software. The intensity of the excitation light and camera settings were carefully tested to ensure no bleed-through and used consistently in the widefield fluorescence imaging (Supplementary Fig. 7). In confocal imaging, the thickness of the optical slice was 0.5 μm. The brightness and contrast of the green and red images were processed by ImageJ and the chromatic shift was corrected by ImageJ plugin Transform J[53] before they were merged as shown in Fig. 2. In analysis of co-localization, Pearson's correlation coefficient was calculated by the Intensity Correlation Analysis plugin of ImageJ for an area containing at least 10 cells as shown in Supplementary Fig. 3 after background subtraction[54].

**Statistics and reproducibility.** All the imaging experiments were repeated at least once at a different time or by a different researcher. All the reported images and data were successfully reproduced. The error bars were derived from at least three independent experiments for the growth curves (Fig. 4), the membrane association bar chart (Supplementary Fig. 5) and activity bar chart (Supplementary Fig. 6).

**Reporting summary.** Further information on research design is available in the Nature Research Reporting Summary linked to this article.

## Data availability
The PlsX structure has been deposited in the Protein Data Bank (PDB ID: 6A1K). All relevant data are available from the corresponding author. The source data underlying Fig. 4 are shown in Supplementary Data 1.

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

## Acknowledgements

This work was supported by National Natural Science Foundation of China (Grant 21877094). We thank the staffs from Shanghai Synchrotron Radiation Facility (SSRF, BL17U1) and National Facility for Protein Science Shanghai (NFPS, BL19U1) for on-site technical assistance during data collection.

## Author contributions

Z.G. and Y.J. conceived and designed the experiments; Y.J. solved the crystal structure; Y.J., X.D., and M.Q. mutated and imaged the fusion protein; Z.G. and Y.J. analyzed and discussed the data; Z.G. and Y.J. wrote the paper.
