## [Peer Review File · Communications Biology]

Reviewers' comments:

Reviewer #1 (Remarks to the Author):

In some bacteria, the plasma membrane does form a single physical phase. In *B. subtilis*, staining with lipid probes adapted to fluid domains shows a dotted pattern. In this study, Yiping Jiang et al present an extensive site-directed mutagenesis of a prominent amphipathic motif at the tip of a four-helix bundle through which the phosphate acyltransferase PlsX in *Bacillus subtilis* dimerizes and probably interacts with the fluid phase of the membrane. The work is based on a new structure of the protein that convincingly shows the amphipathic motif and numerous point mutants, which are analyzed by two-color fluorescence microscopy: one color for the protein and one for the lipid domains. The work is clearly novel and interesting and should be adapted to publication in *Communications Biology* provided that the following points are clarified.

Major point. Many mutations clearly promote protein dissociation from RIFs and lead to a cytosolic – soluble – phenotype. Mutants of this class are those applying to the interfacial amino acids (T253A, T255A and K257A). However, other mutations seem to promote a different phenotype: for example, “the T255W mutant is mostly re- distributed from the RIFs to other parts of the membrane”. I have difficulty at seeing in the images this phenotype and at distinguishing it from the first phenotype. The authors should put effort at illustrating, in an enhanced image, and possibly at quantifying what distinguish these mutants. Clearly the Pearson coefficient is not sufficient. Would it be possible to use a second membrane probe, such as to see the non-RIF phases? Or, alternatively to brake the cell and determine after centrifugation the membrane/soluble partitioning of the mutants? This is a major point because all the conclusions of the authors are based on these contrasting phenotypes, notably the central sentence of the abstract: “Mutations disrupting the amphipathic interaction or INCREASING THE NONPOLAR INTERACTION are found to redistribute the protein to cytosol or OTHER PART OF THE PLASMA MEMBRANE and to cause growth defects”.

Minor points:

#1 In the legend of figure S1 it is written: “with (GFPm) or without (GFP) the A206K mutation “. There is a mistake in the numbering; it should be a value between 250-262.

#2 The authors write: More interestingly, the T255W mutant is mostly re- distributed from the RIFs to other parts of the membrane while the T253W mutant is mainly delocalized to the cytosol (Figure 3). This difference may lie in the fact that Trp255 (in the T255W mutant) is located on the side facing the amphipathic α -peptide of the other subunit in the functional dimer (Figure 1B and 1C) to allow the two symmetric tryptophan residues to interact with each other while Trp253 (in the T253W mutant) is located on the opposite side of the α -peptide and unable to interact similarly. This hypothesis is plausible but not demonstrated. The authors should perform or at least suggest experiment to address this possibility (e.g. looking at the fluorescence property of the protein)

#3 The authors write : “Like the well-known liquid-ordered lipid rafts in both prokaryotic and eukaryotic cells, RIFs are expected to be a platform for organization of proteins”. I would rather write: “Like the well-known liquid-ordered lipid rafts in both prokaryotic and eukaryotic cells but via an opposite physicochemical principle, RIFs are expected to be a platform for organization of proteins”

#4A “majority of each detectable mutant protein was found in the cytosol and eventually distributed throughout the cell”. Please clarify what you mean by “eventually distributed throughout the cell “. Do you mean the non-RIF domains of the membrane?

Reviewer #2 (Remarks to the Author):

In their manuscript, Yiping Jiang and co-authors address the biological function of regions of increased fluidity in bacterial membranes. Those regions were recently identified in two model bacteria, and are believed to consist of short, unsaturated or branched phospholipids and to attract specific proteins. However, no proteins were so far identified that specifically interact with fluid regions of the membrane. In this paper, the authors show that PlsX, an enzyme involved in phospholipid biosynthesis, specifically binds to regions of increased fluidity in the membrane of *Bacillus subtilis* using a short amphipathic helix.

The content of the paper is novel, since this is the first protein known to localise to those micro-domains, thereby providing a biological function to these domains. The experimental work is sound, the data are convincing, and the manuscript is clearly written. To my opinion, the manuscript should be published in *Communications Biology* with only a few minor revisions. First, I find the description of fluorescence microscopy a bit incomplete and at points confusing. It appears that the data obtained from the Nikon microscope are simply referred to as "fluorescence microscopy", as opposed to "confocal fluorescence microscopy". I would recommend the authors to use "widefield fluorescence microscopy" instead, in order to emphasise the difference with the confocal setup. In addition, in the methods section the authors provide the general type of microscope, the objectives and the detection system, but they fail to give any information about the light source used for excitation. I would like to see that information added (lasers, wavelengths, filters, powers, etc.).

And second, it was not clear to me until quite late in the manuscript that most localisation data for mutant PlsX proteins are obtained in the presence of the wild-type protein. On page 7, the authors write "...induced for expression by xylose ... as previously reported". I would prefer that the authors briefly mention that the mutant GFP-tagged proteins are expressed from a plasmid in the presence of the wild-type protein expressed from the genome.

With those clarifications, I believe that this manuscript will contribute to the understanding of membrane organisation in bacteria.

Reviewer #3 (Remarks to the Author):

In this paper the authors have solved the crystal structure of PlsX (phosphate acyltransferase) to find a previously unanticipated dimeric amphipathic peptide of 13 amino acid residues of which 9 form a two and half alpha helix. They have mutated quite a few polar and non-polar residues to residues of opposite polarity to explore the role of this amphipathic peptide in recruiting this protein to RIFs (Regions of increased fluidity) on the membrane. They observe that loss of certain polar residues is enough to lose membrane association, mostly due to loss of hydrogen bonds. However, any alteration in the hydrophobic residues also leads to de-localization of this protein from the membrane. Further, they show that some of these mutations, which lead to delocalization of the protein from RIFs are important in cell growth. There is a noticeable delay in progression to the exponential phase for the mutants.

The novelty of this paper lies mainly in discovery of this amphipathic peptide in this protein, which might indeed be important for its association with RIFs.

However several caveats in the interpretations of the authors abound:

Structure and Function:

- 1) Do the mutations made in the dimeric amphipathic helix influence the structure of the protein, there by influencing all its properties- including association with the RIFs as well as altered cell growth.
- 2) Is the localization to RIFs via this amphipathic peptide important for activity of these proteins? Since the protein in question is important in phospholipid synthesis, the differences in cell growth

might be an effect of loss of its activity. Checking the levels of lipids synthesized in these mutants could elucidate this point and comparing it to a situation when there are no RIFs (eg. MreB deletion condition) would be a useful experiment to conduct.

3) Do the mutations affect the enzymatic activity of the protein due to an influence on its structure (see also 1 above).

Mechanism of interaction with membrane:

4) The experiments do not conclusively show that this protein is indeed interacting directly with the membrane. The mutations might be important in the interaction of the protein with other peripheral proteins, which directly interact with/regulate the RIFs. The amphipathic helix might be necessary in positioning the protein correctly near the membrane to associate with multiple proteins.

5) If it were indeed directly sensing lipids that form fluid regions, the expectation would be increasing polarity of the peptide increases its tendency to associate with the RIFs and increase of hydrophobicity leads to loss of association. However, the decrease of hydrophobicity in the peptide leading to loss of membrane association doesn't quite go with this hypothesis. How do the authors think this fits in the model?

6) Can the authors comment on why they think that this amphipathic helix is not sensing RIFs via mechanisms like sensing or inducing curvature, lipid head group or line tension.

7) The lack of sequence matching among different bacterial genera might imply it is sensing physical properties of the bilayer such as presence of hydrophilic regions or curvature as these regions being made of short chain lipids will be thinner than surrounding regions or line tension rather than binding 'special' lipids. To understand that one could ask if the sequences of these proteins along with the amphipathic peptide have any specific lipid or protein binding head. One could also alter the lipid composition of the cells by altering temperature or gene expression of specific phospholipid synthesis enzymes to see how the RIFs and the localization of PlsX change.

8) Changing the lipid composition in bilayers of varying yet regulated thickness would also help tease apart the following points in in vitro experiments:

- i. Is the protein associating directly with the bilayer
- ii. Is the protein sensing specific lipids or generic bilayer physical properties.

In summary, if the authors indeed wish to make claims about how the protein interacts with the membrane and/ or alters growth of the bacteria, they must shed some light on the questions raised.

Point-to-point responses to Reviewers' comments

Title: Identification of an amphipathic peptide sensor of the bacterial fluid membrane microdomains (ms#COMMSBIO-19-0488-T)

Reviewer #1 (Remarks to the Author):

*In some bacteria, the plasma membrane does form a single physical phase. In *B subtilis*, staining with lipid probes adapted to fluid domains shows a dotted pattern. In this study, Yiping Jiang et al present an extensive site-directed mutagenesis of a prominent amphipathic motif at the tip of a four-helix bundle through which the phosphate acyltransferase PlsX in *Bacillus subtilis* dimerizes and probably interacts with the fluid phase of the membrane. The work is based on a new structure of the protein that convincingly shows the amphipathic motif and numerous point mutants, which are analyzed by two-color fluorescence microscopy: one color for the protein and one for the lipid domains. The work is clearly novel and interesting and should be adapted to publication in *Communications Biology* provided that the following points are clarified.*

Major point. Many mutations clearly promote protein dissociation from RIFs and lead to a cytosolic – soluble – phenotype. Mutants of this class are those applying to the interfacial amino acids (T253A, T255A and K257A). However, other mutations seem to promote a different phenotype: for example, “the T255W mutant is mostly re- distributed from the RIFs to other parts of the membrane”. I have difficulty at seeing in the images this phenotype and at distinguishing it from the first phenotype. The authors should put effort at illustrating, in an enhanced image, and possibly at quantifying what distinguish these mutants. Clearly the Pearson coefficient is not sufficient. Would it be possible to use a second membrane probe, such as to see the non-RIF phases? Or, alternatively to brake the cell and determine after centrifugation the membrane/soluble partitioning of the mutants? This is a major point because all the conclusions of the authors are based on these contrasting phenotypes, notably the central sentence of the abstract: “Mutations disrupting the amphipathic interaction or INCREASING THE NONPOLAR INTERACTION are found to redistribute the protein to cytosol or OTHER PART OF THE PLASMA MEMBRANE and to cause growth defects”.

Response: We appreciate the positive view of the work and agree that T255W is not different from other RIFs-delocalizing mutations in the fluorescent images in Figure 2. T255W was considered to be redistributed to other parts of the membrane rather than being delocalized to cytosol because it was found to be mainly associated with the membrane while other mutants were evenly distributed all over the cells in the confocal images shown in Figure 3. Nevertheless, we followed the suggestion ‘..... to brake the cell and determine after centrifugation the membrane/soluble partitioning of the mutants’ and measured the amount of the GFP_m-fused protein (by fluorescence) in cytosol and membrane after ultracentrifugation (Fig. S5). These results show that T255W is indeed bound to the membrane at a level similar to the wild-type protein, while a significantly lower amount of K257A is bound to the membrane (see below). This provides additional

supporting evidence to our conclusion that T255W is different from other RIFs-delocalizing mutation such as K257A in re-distributing the protein in the membrane rather than delocalizing it to cytosol. The following modifications have been made to the manuscript to add this experimental evidence:

Supporting information: Fig. S5 is added.

Fig.S5. Different partition of PlsX mutants between plasma membrane and cytosol. **a.** The GFP_m-fused protein in cytosol. **b.** The GFP_m-fused protein associated with the membrane. WT: wild-type GFP_m-PlsX; T255W: GFP_m-T255W; and T257A: GFP_m-T257A. *Bacillus subtilis* cells were induced to express the fusion protein at 37°C, grown to OD₆₀₀ = 0.50, harvested, lysed and centrifuged to obtain crude extract, which was ultracentrifuged at 300,000 × g to obtain membrane pellet and supernatant. The pellet was resuspended in PBS buffer with the same volume as the supernatant and the same amount of pellet suspension and supernatant was used in fluorescence measurement with excitation at 488 nm and emission at 508 nm. The experiments were performed in triplicates and the fluorescence was normalized using the result for the wild-type protein. The wild type GFP_m-PlsX is 7.85 fold in cytosol more than in membrane.

Main text: p10, a new sentence (in red) is added as shown below—'.....but the T255W-containing mutant remains associated with the lateral membrane and the midcell region (Figure 3). **In corroboration of this result, the GFP_m-T255W protein was found to be associated with the membrane at a level similar to the wild-type GFP_m-PlsX but significantly higher than GFP_m-T257A (Fig. S5), which was mainly delocalized to cytosol.**'; p15, the relevant sentence is modified—'..... More interestingly, the T255W mutant is mostly re-distributed from the RIFs to other parts of the membrane because it is still associated with the membrane in confocal fluorescence microscopy (Figures 3 and S5), while the T253W mutant is mainly delocalized to the cytosol (Figure 3).'

Materials and Methods: The following paragraph is added to describe how the additional experiment was done in the section 'Expression of GFP_m-PlsX and its mutants in *Bacillus subtilis*.

'To determine the membrane association of proteins, *B. subtilis* cells expressing GFP_m-PlsX, GFP_m-T255W and GFP_m-T257K were individually grown at 37°C as described above to OD₆₀₀ = 0.5. The cells were harvested, resuspended in 10 ml standard PBS buffer (137 mM NaCl, 2.7 mM KCl, 10 mM Na₂HPO₄, and 1.8 mM KH₂PO₄, pH 7.4) and lysed by sonication. Cell debris was removed by centrifugation at 30,000 × g for 15 min, and the supernatant was ultracentrifuged at 100,000 × g for 1 h to separate cell membrane and cytosol. The membrane pellet was then resuspended in PBS buffer supplemented with 1% *n*-dodecyl-β-D-maltopyranoside (DDM). Subsequently, membrane and cytosol fractions were subjected to fluorescence measurement using FlexStation 3 Multi-Mode Microplate Reader (Molecular Devices), GFP fluorescence intensity was measured with excitation at 488±4.5 nm and emission at 507±7.5 nm.'

Minor points:

#1 *In the legend of figure S1 it is written: "with (GFP_m) or without (GFP) the A206K mutation ". There is a mistake in the numbering; it should be a value between 250-262.*

Response: The A206K mutation occurs in GFP and greatly reduces its dimerization tendency. It is not in PlsX and therefore is not in the range 250-262.

#2 *The authors write: More interestingly, the T255W mutant is mostly re-distributed from the RIFs to other parts of the membrane while the T253W mutant is mainly delocalized to the cytosol (Figure 3). This difference may lie in the fact that Trp255 (in the T255W mutant) is located on the side facing the amphipathic α-peptide of the other subunit in the functional dimer (Figure 1B and 1C) to allow the two symmetric tryptophan residues to interact with each other while Trp253 (in the T253W mutant) is located on the opposite side of the α-peptide and unable to interact similarly. This hypothesis is plausible but not demonstrated. The authors should perform or at least suggest experiment to address this possibility (e.g. looking at the fluorescence property of the protein)*

Response: We agree that this is only a speculation. Following the suggestion here, we measured the tryptophan fluorescence of purified T255W mutant (in response to other concerns) but were unable to find change when liposome was added into the solution. This may be due to the fact that most of the protein (~90%) is in the solution even though T255W is still associated with the membrane (Fig. S5). Thus, the fluorescence data is not included in the revised manuscript and we modify the text to explicitly indicate that this is a speculation, not a 'fact' in the original version. The change is shown below with highlight: '...**One possible explanation for this difference is ~~may lie in the fact~~ that Trp255 (in the T255W mutant) is located on the side facing the amphipathic α-peptide of the other subunit in the functional dimer (Figure 1b and 1c) to allow the two**

symmetric tryptophan residues to interact with each other while Trp253 (in the T253W mutant) is located on the opposite side of the α -peptide and unable to interact similarly. However, how this **probable** difference in the side chain interaction leads to different subcellular distribution is not clear.'

#3 The authors write : "Like the well-known liquid-ordered lipid rafts in both prokaryotic and eukaryotic cells, RIFs are expected to be a platform for organization of proteins". I would rather write: "Like the well-known liquid-ordered lipid rafts in both prokaryotic and eukaryotic cells but via an opposite physicochemical principle, RIFs are expected to be a platform for organization of proteins"

Response: We appreciate the suggestion and include the change in the revision, exactly as suggested.

#4 A "majority of each detectable mutant protein was found in the cytosol and eventually distributed throughout the cell". Please clarify what you mean by "eventually distributed throughout the cell ". Do you mean the non-RIF domains of the membrane?

Response: This is a spelling error: 'eventually' should be 'evenly'. This has been corrected. Many thanks for pointing out this error.

Reviewer #2 (Remarks to the Author):

In their manuscript, Yiping Jiang and co-authors address the biological function of regions of increased fluidity in bacterial membranes. Those regions were recently identified in two model bacteria, and are believed to consist of short, unsaturated or branched phospholipids and to attract specific proteins. However, no proteins were so far identified that specifically interact with fluid regions of the membrane. In this paper, the authors show that PlsX, an enzyme involved in phospholipid biosynthesis, specifically binds to regions of increased fluidity in the membrane of Bacillus subtilis using a short amphipathic helix.

The content of the paper is novel, since this is the first protein known to localise to those micro-domains, thereby providing a biological function to these domains. The experimental work is sound, the data are convincing, and the manuscript is clearly written. To my opinion, the manuscript should be published in Communications Biology with only a few minor revisions.

First, I find the description of fluorescence microscopy a bit incomplete and at points confusing. It appears that the data obtained from the Nikon microscope are simply referred to as "fluorescence microscopy", as opposed to "confocal fluorescence microscopy". I would recommend the authors to use "widefield fluorescence microscopy" instead, in order to emphasise the difference with the confocal setup. In addition, in the methods section the authors provide the general type of microscope, the objectives and the detection system, but

they fail to give any information about the light source used for excitation. I would like to see that information added (lasers, wavelengths, filters, powers, etc.).

And second, it was not clear to me until quite late in the manuscript that most localisation data for mutant PlsX proteins are obtained in the presence of the wild-type protein. On page 7, the authors write "...induced for expression by xylose ... as previously reported". I would prefer that the authors briefly mention that the mutant GFP-tagged proteins are expressed from a plasmid in the presence of the wild-type protein expressed from the genome. With those clarifications, I believe that this manuscript will contribute to the understanding of membrane organisation in bacteria.

Response: We appreciate the positive view and the suggestions. As suggested, we have replaced "fluorescence microscopy" with "widefield fluorescence microscopy" throughout the manuscript. For example, changes have been made in p10: "..... Under **widefield** fluorescence microscopy, the A259T-containing mutant was not visible due to poor expression (Fig. S2),....." and "The A261F mutant was not substantially expressed and was invisible under a **widefield** fluorescence microscope,"

We have also specified the light source of the fluorescence microscopes as suggested. The changes are highlighted in the following sentence in the 'Fluorescence imaging' section of 'Materials and Methods': "After the medium dried up due to absorption by the agarose film, the immobilized cells were covered with a coverslip (Marienfeld) and imaged with a Nikon TE2000E-PFS microscope equipped with **an Intensilight Epi-fluorescence Illuminator**, an Andor EMCCD BOOST camera, an Alpha-PL APO 100 × oil objective, **a GFP filter cube, a rhodamine filter cube** and NIS element software or a Zeiss Laser Scanning Confocal Microscope (LSM 710) equipped with a PL APO 63 × oil objective, **488 nm Argon laser, 560 nm DPSS laser source and** Zen 2009 software."

Moreover, we also follow the suggestion to mention that the GFP_m-tagged PlsX is expressed along with the untagged protein expressed from its native locus in the genome. The tagged protein is actually expressed from the *amyE* locus where the plasmid is integrated into the genome after transformation and antibiotic selection. Therefore, the relevant sentence in page 7 is modified as shown below: ".....PlsX was fused to GFP_m at the N-terminus and induced for expression by xylose **from the amyE locus in Bacillus subtilis in the presence of the untagged PlsX expressed from its native locus** exactly as previously reported.³"

Reviewer #3 (Remarks to the Author):

In this paper the authors have solved the crystal structure of PlsX(phosphate acyltransferase) to find a previously unanticipated dimeric amphipathic peptide of 13 amino acid residues of which 9 form a two and half alpha helix. They have mutated quite a few polar and non-polar

residues to residues of opposite polarity to explore the role of this amphipathic peptide in recruiting this protein to RIFs (Regions of increased fluidity) on the membrane. They observe that loss of certain polar residues is enough to lose membrane association, mostly due to loss of hydrogen bonds. However, any alteration in the hydrophobic residues also leads to delocalization of this protein from the membrane. Further, they show that some of these mutations, which lead to delocalization of the protein from RIFs are important in cell growth. There is a noticeable delay in progression to the exponential phase for the mutants. The novelty of this paper lies mainly in discovery of this amphipathic peptide in this protein, which might indeed be important for its association with RIFs.

However several caveats in the interpretations of the authors abound:

Structure and Function:

1) Do the mutations made in the dimeric amphipathic helix influence the structure of the protein, there by influencing all its properties- including association with the RIFs as well as altered cell growth.

2) Is the localization to RIFs via this amphipathic peptide important for activity of these proteins? Since the protein in question is important in phospholipid synthesis, the differences in cell growth might be an effect of loss of its activity. Checking the levels of lipids synthesized in these mutants could elucidate this point and comparing it to a situation when there are no RIFs (eg. MreB deletion condition) would be a useful experiment to conduct.

3) Do the mutations affect the enzymatic activity of the protein due to an influence on its structure (see also 1 above).

Response: We appreciate the positive view and the comments. Points 1) and 3) are actually the same, concerning whether the mutations in the amphipathic helix affect the structure to reduce or eliminate the catalytic activity of PlsX. Point 2) concerns whether the detected growth defect is due to the loss of RIFs localization or loss of the catalytic activity induced by the mutations, and is therefore again a problem about whether the mutations affect the PlsX catalytic activity. Although the mutations were already pointed out to be very far from the active site in the original manuscript, we carried out activity assays to address these concerns. Specifically, we expressed and purified recombinant PlsX and K257A mutant with a N-terminal hexahistidine tag in *E. coli* and determined their catalytic activity. The activity data are added in the supporting information as Fig. S6, which clearly shows no effect for the K257A mutation on the PlsX activity. Since this additional experiment fully address the concerns, we did not compare the levels of lipids between normal cells with RIFs and MreB deletion cells as suggested. Actually, the suggestion that there are no RIFs in the absence of MreB is not in line with the previous finding that RIFs are not affected by knockout of MreB/Mbl/MreBH (in Ref. #1).

The following changes are made in the revision:

Supporting information: Fig. S6 is added.

Fig. S6. Catalytic activity of PlsX and its mutant K257A. a. SDS-PAGE gel of the purified PlsX and its K257A mutant. b. Normalized catalytic activity of the wild-type PlsX and its K257A mutant. See 'Materials and Methods' for the assay conditions of the enzymic activity.

Main text:

A sentence (highlighted in red) is added to the last sentence of the 'Results' section: "Since the point mutations are very far from the suspected active site (Figure 1a) and should not affect the catalytic activity, this growth impairment provides unambiguous evidence for a crucial role for the subcellular localization in the physiological function of PlsX. **This is further supported by the unaffected catalytic activity of the pure recombinant K257A protein in comparison to the non-mutated PlsX (Fig. S6).**"

Materials and Methods: A new section is added to describe the activity assay and lipid binding of PlsX and its K257A mutant. The following is the description of the activity assay in this new section with a new reference (#48, numbering of other references this point onward is changed accordingly)

'Activity assay and direct lipid binding of PlsX and its K257A mutant. To introduce the K257A mutation, the plasmid in pET28 used in PlsX expression was used as a template for site-directed mutagenesis using the primers in Table S1. The K257A mutant was expressed and purified exactly like the wild-type PlsX as described above. The palmitoyl-ACP substrate was prepared enzymatically from palmitoyl-CoA and *holo*-ACP according to a reported method.⁴⁸ The enzymatic activity assay was determined based on coupling with (5, 5-dithio-bis-(2-nitrobenzoic acid) (DTNB), which react with the *holo*-ACP product to yield 2-nitro-5-thiobenzoic acid (TNB) for UV-Vis measurement at 412 with an extinction coefficient of $14,150 \text{ M}^{-1} \cdot \text{cm}^{-1}$. In our activity assay, a 200 μl reaction mixture contained 1 mM MgCl_2 , 500 μM DTNB, 500 μM phosphate, and 10 μM palmitoyl-ACP in 50 mM Tris buffer (pH 7.5). The enzyme was then added to a final concentration of 1 μM and the reaction was monitored for absorbance change at 412 nm a UV-VIS spectrometer (Shimadzu). Assays were performed in triplicate at 25°C.'

Mechanism of interaction with membrane:

4) *The experiments do not conclusively show that this protein is indeed interacting directly with the membrane. The mutations might be important in the interaction of the protein with other peripheral proteins, which directly interact with/regulate the RIFs. The amphipathic helix might be necessary in positioning the protein correctly near the membrane to associate with multiple proteins.*

5) *If it were indeed directly sensing lipids that form fluid regions, the expectation would be increasing polarity of the peptide increases its tendency to associate with the RIFs and increase of hydrophobicity leads to loss of association. However, the decrease of hydrophobicity in the peptide leading to loss of membrane association doesn't quite go with this hypothesis. How do the authors think this fits in the model?*

6) *Can the authors comment on why they think that this amphipathic helix is not sensing RIFs via mechanisms like sensing or inducing curvature, lipid head group or line tension.*

7) *The lack of sequence matching among different bacterial genera might imply it is sensing physical properties of the bilayer such as presence of hydrophilic regions or curvature as these regions being made of short chain lipids will be thinner than surrounding regions or line tension rather than binding 'special' lipids. To understand that one could ask if the sequences of these proteins along with the amphipathic peptide have any specific lipid or protein binding head. One could also alter the lipid composition of the cells by altering temperature or gene expression of specific phospholipid synthesis enzymes to see how the RIFs and the localization of PlsX change.*

8) *Changing the lipid composition in bilayers of varying yet regulated thickness would also help tease apart the following points in in vitro experiments:*

i. Is the protein associating directly with the bilayer

ii. Is the protein sensing specific lipids or generic bilayer physical properties.

In summary, if the authors indeed wish to make claims about how the protein interacts with the membrane and/ or alters growth of the bacteria, they must shed some light on the questions raised.

Response: Due to the lack of experimental evidence for direct interaction of PlsX with the membrane, alternative models are suggested for PlsX's membrane association and a number of experiments are suggested for their test. To address this concern, we prepared liposomes with protein-free total lipids from *Bacillus subtilis* and incubated them with recombinant PlsX, T257A and lysozyme as the negative control. After ultracentrifugation, the liposomes were collected for SDS-PAGE. The results are shown in an added figure, Fig. S4, in the supporting information, which clearly show that PlsX binds directly to the membrane and the binding is significantly weakened by the T257A

mutation. This direct evidence makes the suggested alternative models and testing experiments unnecessary. More responses are provided below to specific points.

Point 4): the newly collected direct evidence answers fully the concern expressed here.

Point 5): As specified in the manuscript, the hydrophobic face of the amphipathic peptide interacts with the acyl part of the membrane lipids while its interfacial residues interact with the polar head groups. Thus, when hydrophobicity is reduced in the nonpolar face of the helix, binding with the membrane is reduced to result in loss of membrane association. This is a general phenomenon among the membrane association mediated by all amphipathic peptides, not just the one we studied in this work.

Point 6): RIFs are not reported to involve curvature induction or possess special lipid head group or line tension (please see Ref. #1 and #3) as mentioned here. Actually, their only characteristic is the increased fluidity. This is why we did not explain the results in terms of the mentioned physicochemical properties.

Points 7) and 8): Currently, DesK is the only known protein to sense membrane fluidity by responding to the membrane thickness. It is, however, a multipass membrane protein that depends on the transmembrane helices to sense the perturbation in thickness. Since PlsX has been shown to bind the membrane directly (Fig. S4) and to interact with membrane peripherally, it lacks the necessary transmembrane helix to sense the thickness. In addition, the suggested alternative model of sensing hydrophobicity or curvature as a result of the perceive implication of the lack of sequence conservation still cannot explain why a non-conserved motif should sense a conserved feature of the fluid microdomains. In contrast, the general difference in lipids between *Bacillus* and other bacterial genera is a more likely cause. This is because, as pointed out in the manuscript, *Bacillus* is different from all other bacteria in having overwhelmingly saturated fatty acyls with either *iso*- or *anteiso*-branches.

The suggestion to change lipid composition to alter RIFs is not yet realizable at present due to our limited understanding of lipids in RIFs. Although RIFs are suggested to compose of unsaturated, short/branched fatty acyl lipids (Ref. #1 & 3), the unsaturated fatty acyl groups are negligible and all fatty acyl groups are either *iso*- or *anteiso*-branched in *Bacillus subtilis* (Ref. #31 & 32) as pointed out in our manuscript. In addition, the length distribution of fatty acyl in the lipids is controlled by the intrinsic catalytic properties of the fatty acid synthase and is not known to be changeable. Thus, to change RIFs via lipid composition requires better understanding of the fluid microdomains.

The changes made in the revision are detailed below:
Supporting information: Fig. S4 is added.

Fig.S4. Direct binding of PlsX to protein-free lipids of *Bacillus subtilis*. As a negative control, lysozyme was negligibly bound to the liposomes, while PlsX was significantly bound to the lipids. When the interaction was impaired by the K257A mutation, the amount of protein bound to the liposomes greatly decreased. These results provide strong support for the direct binding interaction between PlsX and the lipids. Protein-free total lipids were isolated from *Bacillus subtilis* and used to make liposome as described in ‘Materials and Methods’. The tested proteins were incubated with the protein-free liposomes at a concentration of 2 mg/ml for 1 h at room temperature. The liposomes were separated from the supernatant by ultracentrifugation at $200,000 \times g$ and re-suspended in buffer for analysis by SDS-PAGE.

Main text:

A sentence (highlighted in red) is added to the end of ‘*Amphipathicity required for the RIFs localization*’ section: “**Moreover, recombinant PlsX was found to directly bind to protein-free total lipids from *Bacillus subtilis* and the K257A mutation was found to significantly weaken this interaction (Fig. S4).** Taken together, these results provide strong evidence that PlsX interacts directly with membrane for its peripheral association with RIFs and that the amphipathic α -peptide is responsible for this subcellular localization.”

Materials and Methods: A paragraph is added to describe the lipid binding experiment in the new section ‘**Activity assay and direct lipid binding of PlsX and its K257A mutant**’:

“The protein-free total lipids were prepared from *Bacillus subtilis* using a reported method.³² Briefly, *Bacillus subtilis* 168 strain was grown in LB medium under aerobic condition at 37°C and harvested when OD₆₀₀ reached 0.5. Cells from 2 L liquid culture were combined and washed twice with 1% NaCl solution. Subsequently, the cell pellet was added 8 ml chloroform, 16 ml methanol and 6 ml water and extracted by vigorous mixing. The resulting mixture was left at 25°C overnight, followed by addition of 8 ml chloroform and 8 ml water. The lower phase was collected and solvent was removed by rotary evaporation to obtain a lipid film. Then the lipid film was resuspended in 2 ml buffer containing 10 mM HEPES (pH 7.5) and 150 mM NaCl and liposome was prepared by 1 h sonication treatment. Subsequently, wild-type PlsX, K257A and lysozyme were added at a final concentration of 2 mg/ml to equal aliquots of liposome and incubated with at 25°C for 1 h.

The mixtures were then ultracentrifuged at $200,000 \times g$ for 1 h to collect the liposome fraction for analysis by SDS-PAGE.”

REVIEWERS' COMMENTS:

Reviewer #1 (Remarks to the Author):

the authors have addressed most of my points by additional experiments or textual changes. I recommend publication of this interesting ms.

minor point : correct enzymic into enzymatic in fig s6

Reviewer #2 (Remarks to the Author):

The previous concerns have been carefully addressed by the authors, and the additional data make the manuscript even more convincing. I recommend this manuscript to be published in Communications Biology.

Reviewer #3 (Remarks to the Author):

The authors have addressed all the substantial parts of the issues raised, and the new experimental data concerning the catalytic activity as well as direct lipid association of the Ptx protein, provide convincing evidence for the main thesis of the author. This relates to the mechanism of localization of proteins to RIFs and the control of their function by this localization. While this study opens up further mechanistic questions, it certainly provides convincing evidence for the claims made therein.